# High-molecular weight DNA extraction, clean-up and size selection for long-read sequencing

**Ashley Jones**[1]*, **Cynthia Torkel**[1], **David Stanley**[1,2], **Jamila Nasim**[1,3], **Justin Borevitz**[1], **Benjamin Schwessinger**[1]

**1** Research School of Biology, Australian National University, Canberra, Australian Capital Territory, Australia, **2** Diversity Arrays Technology, Bruce, Australian Capital Territory, Australia, **3** Soil Carbon Co., Orange, New South Wales, Australia

* ashley.jones@anu.edu.au

**Data Availability Statement:** Sequencing data and reference genomes generated with this protocol are being made publicly available on the Sequence Read Archive (SRA, NCBI) under BioProjects PRJNA509734 and PRJNA510265. https://www.ncbi.nlm.nih.gov/bioproject/509734 https://www.

## Abstract

Rapid advancements in long-read sequencing technologies have transformed read lengths from bps to Mbps, which has enabled chromosome-scale genome assemblies. However, read lengths are now becoming limited by the extraction of pure high-molecular weight DNA suitable for long-read sequencing, which is particularly challenging in plants and fungi. To overcome this, we present a protocol collection; high-molecular weight DNA extraction, clean-up and size selection for long-read sequencing. We optimised a gentle magnetic bead based high-molecular weight DNA extraction, which is presented here in detail. The protocol circumvents spin columns and high-centrifugation, to limit DNA fragmentation. The protocol is scalable based on tissue input, which can be used on many species of plants, fungi, reptiles and bacteria. It is also cost effective compared to kit-based protocols and hence applicable at scale in low resource settings. An optional sorbitol wash is listed and is highly recommended for plant and fungal tissues. To further remove any remaining contaminants such as phenols and polysaccharides, optional DNA clean-up and size selection strategies are given. This protocol collection is suitable for all common long-read sequencing platforms, such as technologies offered by PacBio and Oxford Nanopore. Using these protocols, sequencing on the Oxford Nanopore MinION can achieve read length N50 values of 30–50 kb, with reads exceeding 200 kb and outputs ranging from 15–30 Gbp. This has been routinely achieved with various plant, fungi, animal and bacteria samples.

## Introduction

DNA sequencing technologies have transformed genomics due to rapid advances in read length, throughput and application, combined with an ever competitive price. Short-read sequencing platforms provide billions of reads 100–250 bp in length at unrivalled accuracy, while long-read platforms a can provide millions of reads 1 kbp to 1 Mbp, at the cost of accuracy [1]. Long-read platforms have been at the forefront of recent advancements, as they offer

ncbi.nlm.nih.gov/sra?linkname=bioproject_sra_all&from_uid=510265.

**Funding:** A.J. and B.S. received sequencing funds from Bioplatforms Australia, as part of the Genomics for Australian Pants initiative www.genomicsforaustralianplants.com J.B. received funds from an Australian Research Council Centre of Excellence (Plant Energy Biology) (CE140100008) and Discovery Project (DP150103591) www.arc.gov.au B.S. received funds from an Australian Research Council Future Fellowship (FT180100024) www.arc.gov.au The funders had and will not have a role in study design, data collection and analysis, decision to publish, or preparation of the manuscript.

**Competing interests:** "For two plant species, Wahlenbergia ceracea and Phebalium stellatum, sequencing funds have been generously provided by Bioplatforms Australia, under the Genomics for Australian Plants initiative. The funder had no influence on the interpretation of the data." This does not alter our adherence to PLOS ONE policies on sharing data and materials. Many datasets are publicly available on NCBI, others will become available in due time with publication of the genome assembly. There are no competing interests relating to employment, consultancy, patents, products in development, or marketed products. We present a full and objective manuscript that has no interference to its objectivity.

unprecedented opportunities for *de novo* assembly of full length chromosomes and phasing of haplotypes [2,3]. With long-read sequencing platforms, advancements have shifted read length being limited by technology to being limited by quality and length of the DNA input. This has given rise to a new challenge; the extraction of pure high-molecular weight DNA suitable for long-read sequencing, which is particularly troublesome in plants and fungi. This is often caused by the presence of secondary metabolites such as polyphenols and polysaccharides. Polyphenols within the cytosol will be exposed to DNA after cell lysis and can have irreversible interactions [4]. Polysaccharides can co-precipitate with DNA in the presence of alcohol and can have downstream inhibitory effects in many molecular biology techniques [5]. Isolation of nuclei can help resolve these issues and obtain high-molecular weight DNA [6]. Indeed, nuclei preps have been further developed for long-read sequencing but remain laborious and low throughput [7]. One approach that is becoming widely utilized for long-read sequencing is the use of carboxylated magnetic beads, which DNA can bind to under the presence of polyethylene glycol and sodium chloride [8]. This method does not isolate nuclei but still circumvents the use of binding columns and high centrifugation, which are techniques that can fragment DNA. Here we present a modified protocol of Mayjonade et al. [8] that has been used across a wide variety genera of samples, including recalcitrant plants. For plants containing excessive phenols and polysaccharides, an optional washing of homogenate with sorbitol is included which help remove these contaminants [9]. Lastly, DNA clean-up and size selection options are presented which can greatly enhance the success of long-read sequencing platforms. This protocol is part of a bigger repository hosted on Protocols.io, as part of the public workspace 'High-molecular weight DNA extraction from all kingdoms' (https://www.protocols.io/workspaces/high-molecular-weight-dna-extraction-from-all-kingdoms).

## Methods

The protocol described in this article is published on protocols.io, https://dx.doi.org/10.17504/protocols.io.bss7nehn.

## Expected results

Using the protocol described, we have been obtaining large yields of high-molecular weight DNA (Table 1, Fig 1). DNA fragment size ranges from 20–200 kb in length, which is ideal for long-read sequencing (Fig 1). To remove the small DNA fragments and clean plant DNA preps which can be somewhat crude, PippinHT (Sage Science) to select fragments 20 kb and above has been very efficient (Table 1). Other DNA clean-up options are presented in the protocol and achieve similar results, however are more labour intensive. During sequencing, we can reproducibly obtain over 15–30 Gbp of reads from a single Oxford Nanopore MinION flow cell, with read length N50s 30–50 kb (Table 2, Fig 2). This includes quality reads over 200 kb in length (> Q7, Phred scale). It is likely smaller fragments are favoured during sequencing (higher molarity) and the library prep is likely to cause some DNA shearing. Sequencing with PacBio Sequel II (circular consensus sequencing mode for HiFi reads), yields over 20 Gbp can be achieved at very high accuracy (> Q30), but at a smaller length, as this technology is optimised for 10–20 kb fragments. High performing sequencing results have been achieved with various plant, fungi, animal and bacteria samples (Table 2). The sequencing data is being used for *de novo* genome assemblies and in some instances haplotype phasing. Sequencing data and the subsequent reference genomes being generated in this project are being made publicly available Sequence Read Archive (SRA, NCBI). Multiple Eucalyptus genomic datasets are available under BioProject PRJNA509734 and Acacia

**Table 1. Fluorometer and Spectrophotometer results of a DNA extraction for *Eucalyptus caleyi*.** Firstly, crude DNA was extracted, which was then size selected for 20 kb and above with a PippinHT (Sage Science).

| Sample | Input | Yield µg | Qubit ng/µL | Nanodrop ng/µL | 260/280 | 260/230 |
|---|---|---|---|---|---|---|
| *E. caleyi* **Leaf to crude DNA** | 10 g leaves (fresh) | 59.60 | 298 | 564 | 1.48 | 0.72 |
| *E. caleyi* **Pippin Prep 20 kb** | 30 µg crude DNA | 13.68 | 228 | 283 | 1.91 | 2.38 |

Readings taken with a Qubit 2.0 Qubit fluorometer (Thermo Fisher Scientific) and a Nanodrop 1000 (Thermo Fisher Scientific).

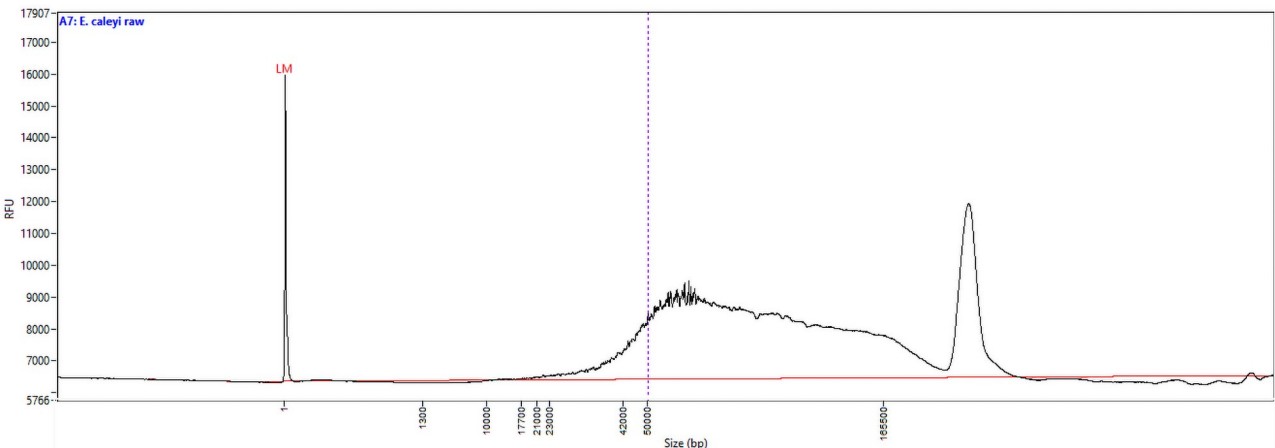

**Fig 1. DNA size distribution of *Eucalyptus caleyi* based on a pulsed-field capillary electrophoresis system, a Femto Pulse (Agilent).** Peak at 200 kb represents all fragments > 200 kb, as they cannot be resolved with the technology. Sample was crude DNA prior to and size selection or further DNA clean-up.

**Table 2. Example long-read sequencing results for multiple samples using different sequencing platforms.**

| Sample | Kingdom | Size selection | Platform | Library prep | Flow cell | Output (Gbp) | N50 Read length (kb) | Mean read length (kb) | Median read length (kb) | Mean read quality (Phred) | Median read quality (Phred) | Longest read (kb) | Longest read ≥ Q7 (kb) |
|---|---|---|---|---|---|---|---|---|---|---|---|---|---|
| Eucalyptus caleyi | Plantae | PippinHT (≥ 20 kb) | ONT MinION | SQK-LSK109 | FLO-MIN106D R9.4.1 | 25.57 | 38.99 | 26.04 | 22.68 | Q10.9 | Q11.50 | 245,735 (Q11.1) | 245,735 (Q11.1) |
| Acacia melanoxylon | Plantae | PippinHT (≥ 20 kb) | ONT MinION | SQK-LSK109 | FLO-MIN106D R9.4.1 | 20.78 | 43.07 | 29.33 | 25.00 | Q9.8 | Q10.3 | 304,552 (Q7.4) | 304,552 (Q7.4) |
| Oryza australiensis | Plantae | PippinHT (≥ 20 kb) | ONT MinION | SQK-LSK109 | FLO-MIN106D R9.4.1 | 15.40 | 44.86 | 27.14 | 21.34 | Q9.7 | Q10.4 | 321,653 (Q8.6) | 321,653 (Q8.6) |
| Triticum aestivum | Plantae | SRE XS (≥ 10 kb) | ONT MinION | SQK-LSK109 | FLO-MIN106D R9.4.1 | 33.21* | 12.42 | 8.19 | 6.12 | Q10.8 | Q11.3 | 12,977 (Q7.8) | 12,977 (Q7.8) |
| Nannizziopsis barbatae | Fungi | PippinHT (≥ 20 kb) | ONT MinION | SQK-LSK109 | FLO-MIN106D R9.4.1 | 3.46† | 32.95 | 23.74 | 22.62 | Q11.3 | Q12.2 | 182,089 (Q9.1) | 182,089 (Q9.1) |
| Puccinia graminis germinated spores | Fungi | SRE XS (≥ 10 kb) | ONT MinION | SQK-LSK109 | FLO-MIN106D R9.4.1 | 7.82† | 28.86 | 17.11 | 12.72 | Q11.8 | Q12.6 | 464,014 (Q4.0) | 145,682 (Q9.6) |
| Gehyra lapistola | Animalia | SRE (≥ 25 kb) | ONT MinION | SQK-LSK109 | FLO-MIN106D R9.4.1 | 9.22 | 16.16 | 8.58 | 5.45 | Q10.5 | Q11.2 | 196,384 (Q3.2) | 125,703 (Q7.7) |
| Escherichia coli BACS (multiplexed) | Bacteria | SRE (≥ 25 kb) | ONT MinION | SQK-LSK109 | FLO-MIN106D R9.4.1 | 21.41 | 20.54 | 13.00 | 9.99 | Q12.2 | Q13.1 | 162,741 (Q2.9) | 147,076 (Q7.9) |
| Phebalium stellatum | Plantae | PippinHT (≥ 20 kb) | ONT PromethION | SQK-LSK110 | FLO-PRO002 | 132.39 | 44.75 | 25.85 | 20.62 | Q8.1 | Q8.3 | 1,109,097 (Q4.6) | 352,235 (Q8.5) |
| Wahlenbergia ceracea | Plantae | SageELF (~13 kb) | PacBio Sequell II HiFi | SMRTbell Express 2.0 | SMRTcell 8M | 23,91 | 12.20 | 12.19 | 12.13 | Q33.5 | Q33.6 | 34,652 (Q21.3) | 34,652 (Q21.3) |

*DNA sheared with 29 gauge needle before size selection to optimise sequencing output.

†Flow cell could be run longer, including flow cell washes, to achieve more output. For small genomes flow cell was stopped when sufficient coverage was achieved.

DNA was isolated using the protocol presented and a final size selection was performed, using either a PippinHT prep (Sage Science), a SageELF (Sage Science) or Short Read Eliminator (SRE) (Circulomics). Sequencing was performed with either Oxford Nanopore Technologies (ONT) MinION, ONT PromethION or PacBio Sequel II in circular consensus sequencing (CCS) mode for HiFi reads. Total sequencing output, length N50, median quality (Phred scale) and longest reads are shown. ONT fast5 data was basecalled post-sequencing with the manufacturer's algorithms provided in Guppy version 4.4.1 (ONT).

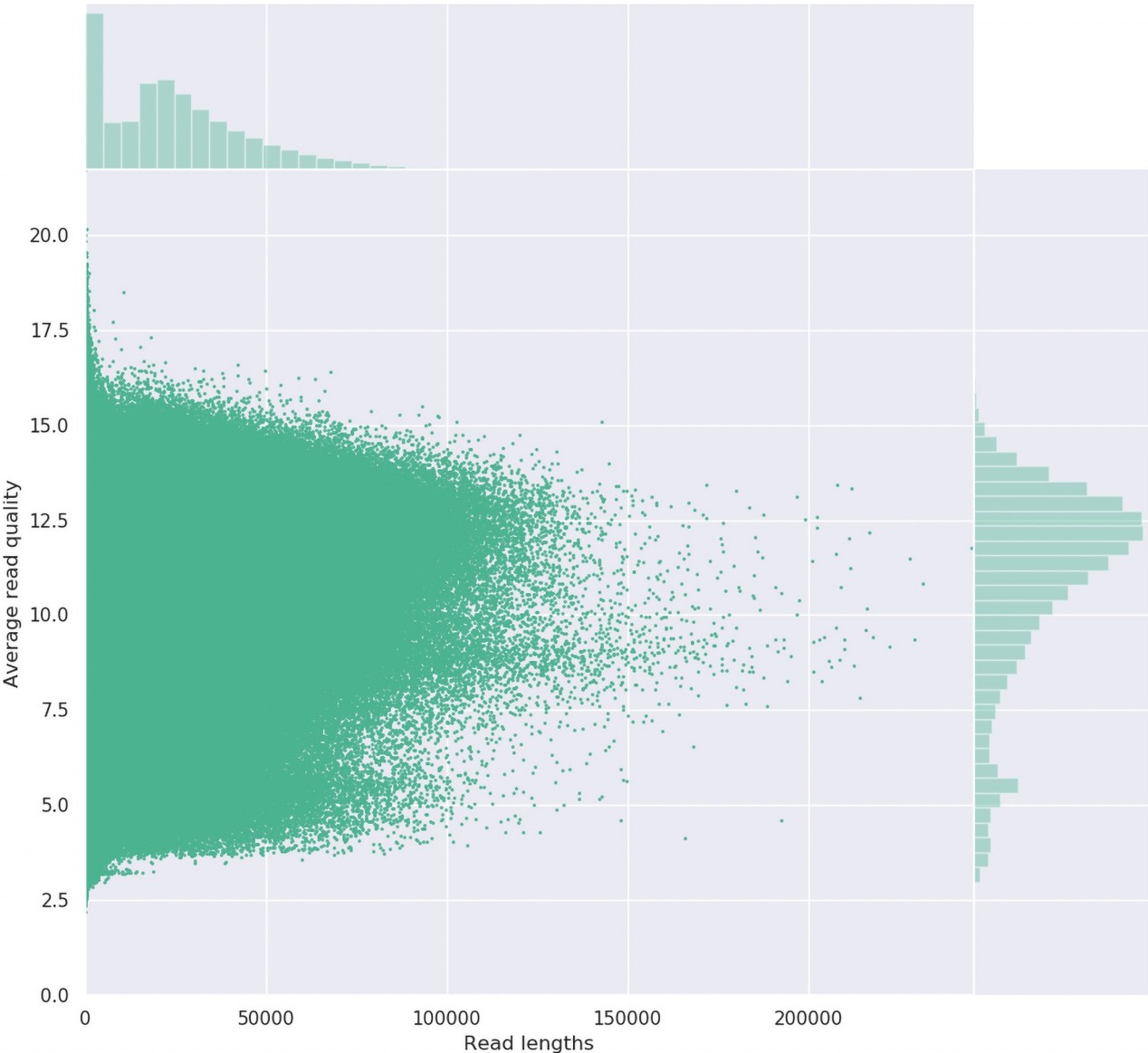

**Fig 2. Read length by average read quality for *Eucalyptus caleyi* long-read sequencing with an Oxford Nanopore MinION flow cell.** Image generated with NanoPlot 1.28.2 [10].

datasets are available under BioProject PRJNA510265. Supporting publications and other genera are soon to follow.

## Supporting information

**S1 Protocol collection. Step-by-step protocol, also available on protocols.io.**
(PDF)

## Author Contributions

**Conceptualization:** Ashley Jones, Justin Borevitz, Benjamin Schwessinger.

**Data curation:** Ashley Jones, Cynthia Torkel, David Stanley, Jamila Nasim, Justin Borevitz, Benjamin Schwessinger.

**Formal analysis:** Ashley Jones, Justin Borevitz, Benjamin Schwessinger.

**Funding acquisition:** Ashley Jones, Justin Borevitz, Benjamin Schwessinger.

**Investigation:** Ashley Jones, Justin Borevitz, Benjamin Schwessinger.

**Methodology:** Ashley Jones, Cynthia Torkel, David Stanley, Jamila Nasim, Justin Borevitz, Benjamin Schwessinger.

**Project administration:** Ashley Jones, Cynthia Torkel, David Stanley, Jamila Nasim, Justin Borevitz, Benjamin Schwessinger.

**Resources:** Ashley Jones, Justin Borevitz, Benjamin Schwessinger.

**Supervision:** Ashley Jones, Justin Borevitz, Benjamin Schwessinger.

**Validation:** Ashley Jones, Cynthia Torkel, David Stanley, Jamila Nasim, Justin Borevitz, Benjamin Schwessinger.

**Visualization:** Ashley Jones, Justin Borevitz, Benjamin Schwessinger.

**Writing – original draft:** Ashley Jones.

**Writing – review & editing:** Ashley Jones, Benjamin Schwessinger.

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
