## [Decision Letter · Decision Letter 0]

27 Apr 2021

PONE-D-21-06732

High-molecular weight DNA extraction, clean-up and size selection for long-read sequencing

PLOS ONE

Dear Dr. Jones,

Thank you for submitting your manuscript to PLOS ONE. After careful consideration, we feel that it has merit but does not fully meet PLOS ONE’s publication criteria as it currently stands. Therefore, we invite you to submit a revised version of the manuscript that addresses the points raised during the review process.

We look forward to receiving your revised manuscript.

Kind regards,

Mark Eppinger

Academic Editor

PLOS ONE

Journal Requirements:

3)  Thank you for stating the following in the Competing Interests section:

[The authors have declared that no competing interests exist.].

We note that you received funding from a commercial source: Bioplatforms Australia

Reviewers' comments:

Reviewer's Responses to Questions

**Comments to the Author**

1. Does the manuscript report a protocol which is of utility to the research community and adds value to the published literature?

Reviewer #1: Yes

2. Has the protocol been described in sufficient detail?

Descriptions of methods and reagents contained in the step-by-step protocol should be reported in sufficient detail for another researcher to reproduce all experiments and analyses. The protocol should describe the appropriate controls, sample sizes and replication needed to ensure that the data are robust and reproducible.

Reviewer #1: Yes

3. Does the protocol describe a validated method?

Protocols should already have been demonstrated to work in the published literature. There should be at least one original research article referenced in the manuscript in which the protocol was used to generate data.

Reviewer #1: Yes

4. If the manuscript contains new data, have the authors made this data fully available?

Reviewer #1: Yes

**5. Is the article presented in an intelligible fashion and written in standard English?**

Reviewer #1: Yes

6. Review Comments to the Author

Reviewer #1: In the manuscript titled as “High-molecular weight DNA extraction, clean-up and size selection for long-read sequencing” Jones et al present a lab protocol for the isolation of high molecular weight DNA from plant tissue. Since Oxford Nanopore Technology (ONT) sequencing is highly sensitive to the DNA quality, this protocol would be very useful to the plant community. Furthermore, the authors claim 15-30 Gbp of sequencing data from a single MinION flow-cell. I find it very impressive and would like to congratulate the authors on achieving that. Overall, the protocol is clearly written and well elaborated. However, I have some minor suggestions for the authors which could further improve the quality of the manuscript.

Line 26: the authors claim that the protocol works for both ONT and PacBio sequencing whereas they only present the data for ONT.

Line 30: “eucalypts” is wrongly spelled.

Line 30: The authors claim that this DNA isolation protocol has been used for several organisms such as eucalyptus, acacias, rice, themeda, wheat, various fungi, 31 geckos, skinks, ticks, ladybird beetles, caterpillars and E. coli but there is no literature reference pointing to the publications where this protocol was used for the above-mentioned species.

Line 84: “includes quality reads over”. Please mention the q-score for referring a read as high quality.

Table 1: The authors should mention if 10 gm was fresh wait or powdered tissue.

Table 2: I would also like to see the mean and the median read length together with N50 in this table. Also, the authors should state the Guppy (or Bonito) version used for base-calling.

7. PLOS authors have the option to publish the peer review history of their article (what does this mean?). If published, this will include your full peer review and any attached files.

Reviewer #1: **Yes: **Harmeet Singh Chawla

---

## [Author Response · Author response to Decision Letter 0]

3 Jun 2021

We thank the editor and reviewers for the thoughtful feedback. We now have addressed all outstanding concerns. We also provide a point-by-point reply to all the reviewers comments. 

In accordance with your specific recent review request, we have also included updated competing interests statements in the cover letter. Thank you

---

## [Editor Report · Decision Letter 1]

14 Jun 2021

High-molecular weight DNA extraction, clean-up and size selection for long-read sequencing

PONE-D-21-06732R1

Dear Dr. Jones,

We’re pleased to inform you that your manuscript has been judged scientifically suitable for publication and will be formally accepted for publication once it meets all outstanding technical requirements.

Kind regards,

Mark Eppinger

Academic Editor

PLOS ONE
---

## [Editor Report · Acceptance letter]

5 Jul 2021

PONE-D-21-06732R1 

High-molecular weight DNA extraction, clean-up and size selection for long-read sequencing 

Dear Dr. Jones:

I'm pleased to inform you that your manuscript has been deemed suitable for publication in PLOS ONE. Congratulations! Your manuscript is now with our production department. 

Kind regards, 

on behalf of

Dr. Mark Eppinger 

Academic Editor

PLOS ONE